# Establishment and Characterization of a Murine Mucosal Mast Cell Culture Model

**DOI:** 10.3390/ijms21010236

**Published:** 2019-12-29

**Authors:** Aya Kakinoki, Tsuyoshi Kameo, Shoko Yamashita, Kazuyuki Furuta, Satoshi Tanaka

**Affiliations:** 1Department of Immunobiology, Faculty of Pharmacy and Pharmaceutical Sciences, Okayama University, Tsushima naka 1-1-1, Kita-ku, Okayama 700-8530, Japan; 2Department of Immunobiology, Okayama University Graduate School of Medicine, Dentistry, and Pharmaceutical Sciences, Tsushima naka 1-1-1, Kita-ku, Okayama 700-8530, Japanfurutak@okayama-u.ac.jp (K.F.); 3Department of Pharmacology, Division of Pathological Sciences, Kyoto Pharmaceutical University, Misasagi Nakauchi-cho 5, Yamashina-ku, Kyoto 607-8414, Japan

**Keywords:** mast cell, IL-9, chymase, histamine, ATP

## Abstract

Accumulating evidence suggests that mast cells play critical roles in disruption and maintenance of intestinal homeostasis, although it remains unknown how they affect the local microenvironment. Interleukin-9 (IL-9) was found to play critical roles in intestinal mast cell accumulation induced in various pathological conditions, such as parasite infection and oral allergen-induced anaphylaxis. Newly recruited intestinal mast cells trigger inflammatory responses and damage epithelial integrity through release of a wide variety of mediators including mast cell proteases. We established a novel culture model (IL-9-modified mast cells, MCs/IL-9), in which murine IL-3-dependent bone-marrow-derived cultured mast cells (BMMCs) were further cultured in the presence of stem cell factor and IL-9. In MCs/IL-9, drastic upregulation of *Mcpt1* and *Mcpt2* was found. Although histamine storage and tryptase activity were significantly downregulated in the presence of SCF and IL-9, this was entirely reversed when mast cells were cocultured with a murine fibroblastic cell line, Swiss 3T3. MCs/IL-9 underwent degranulation upon IgE-mediated antigen stimulation, which was found to less sensitive to lower concentrations of IgE in comparison with BMMCs. This model might be useful for investigation of the spatiotemporal changes of newly recruited intestinal mast cells.

## 1. Introduction

Intestinal and respiratory tracts are constantly exposed to a wide variety of stimulatory factors and are in a continuous state of change. Immune cells located at these tracts have to adapt themselves to these changes in a timely manner. Mast cells are one of the most suitable immune cells that regulate immune responses at such interfaces because they express a diverse array of surface receptors and undergo flexible local transdifferentiation [1,2]. They were found not only to trigger inflammatory responses but also to be involved in immune suppression [3]. Accumulating evidence suggests that intestinal mast cells play critical roles in parasite expulsion through their multiple functions, including regulation of epithelial and endothelial functions and modulation of innate and adaptive immunity [4,5,6].

Interleukin-9 (IL-9), which was also found to be one of the critical mediators for worm expulsion, was first identified as a mast cell growth factor [7]. IL-9 transgenic mice exhibited rapid expulsion of the intestinal nematodes and local mastocytosis in epithelial layer of the gut, trachea and kidney [8,9,10]. The number of intestinal mast cells were unchanged in the gene-targeted mice lacking IL-9 or IL-9 receptor-α chain, whereas oral-antigen-induced accumulation of intestinal mast cells was impaired in these mice [11,12]. These findings indicate that IL-9 should trigger intestinal mastocytosis upon parasite infection and newly recruited mast cells should make a significant contribution towards worm expulsion. IL-9 was found to enhance stem cell factor (SCF)-mediated proliferation of murine bone-marrow-derived cultured mast cells, although IL-9 alone could not support mast cell growth and survival [13]. It remains largely unknown how transdifferentiation of intestinal mast cells should occur under the influence of IL-9 and SCF.

IL-3-dependent bone-marrow-derived cultured mast cells (BMMCs) have been investigated as a useful model of murine immature mast cells. BMMCs were initially regarded as a model of mucosal mast cells, because they stored chondroitin sulphate E, rather than heparin, in their granules and had lower amounts of histamine, both being characteristic of persistent mucosal mast cells [14]. However, accumulating evidence has suggested that there are many differences among BMMCs, resident mucosal mast cells and recruited mast cells upon parasite infection. IL-3 was found to play critical roles in parasite expulsion and intestinal mastocytosis [15]. Murine mucosal mast cells recruited upon parasite infection were found to highly express a series of chymase genes, such as *Mcpt1* [16], although little or no expression of these genes was confirmed in BMMCs. Furthermore, because the number of mucosal resident mast cells is quite small, they remain to be fully characterized. Because BMMCs are highly capable for further differentiation into mature mast cells, local reconstitution of BMMCs in recently developed mast-cell-deficient mice has been used as one of the best suitable approaches to clarify the functions of tissue mast cells [17]. We previously established a modified coculture method of BMMCs using murine fibroblastic cell line, Swiss 3T3, which shared many characteristics with murine cutaneous mast cells [18,19]. We tried to develop here a novel culture model, in which BMMCs were further cultured in the presence of IL-9 and SCF. This model at least partly reflected the characteristics of intestinally recruited mast cells and provided some insights into the process of transdifferentiation of newly recruited mast cells in intestinal tissues.

## 2. Results

### 2.1. Combination of IL-9 and SCF Induced Expression of Mcpt1 and Mcpt2 and Depleted Histamine in Murine BMMCs

Accumulating evidence suggests that SCF plays critical roles in growth and survival of murine tissue mast cells, which are enhanced by IL-9 in mucosal tissues. BMMCs, which are regarded as an immature mast cell population, were found to be obtained when murine bone marrow cells were cultured for about one month in the presence of IL-3. They have potential to undergo further differentiation in response to the environment changes. We first investigated the effects of IL-9 on BMMCs and found that IL-9 alone or in combination with IL-3 could not support the further survival of BMMCs. We, therefore, added SCF, which is responsible for growth and maturation of the connective-tissue-type mast cells and is also abundantly expressed in the intestinal tissues, to the culture to support survival of the cultured mast cells. Expression of *Mcpt1* and *Mcpt2*, both regarded as characteristic of mucosal mast cells [13], was drastically induced during the culture period in the presence of SCF and IL-9, whereas that of *Mcpt5* and *Cpa3* was upregulated and maintained (Figure 1a–d). It was noteworthy that a drastic downregulation of *Hdc*, which encodes the enzyme responsible for histamine synthesis, was unexpectedly observed (Figure 1e). We previously demonstrated that *Ptgr1*, which encodes the enzyme involved in inactivation of the eicosanoids, was expressed in the mucosal mast cells, not in the connective-tissue-type mast cells in murine stomach [20]. Expression of *Ptgr1* was found to be moderately upregulated in our system (Figure 1f). Surface expression levels of FcεRI and c-kit were significantly decreased in the presence of SCF and IL-9 (Figure 1g,h).

We then measured the enzymatic activities of the cultured mast cells. Chymotryptic activity was significantly increased in BMMCs cultured in the presence of SCF and IL-9 for 12 days (designed as IL-9-modified mast cells, MCs/IL-9), whereas tryptic activity was downmodulated (Figure 2a,b). Carboxypeptidase A activities varied greatly in MCs/IL-9 (Figure 2c). In agreement with the expression levels of *Hdc*, histamine synthesis was drastically suppressed in the presence of SCF and IL-9 (Figure 2d,e). Because tissue mast cells are often located next to cells expressing the membrane-bound form of SCF, we performed the coculture with a murine fibroblastic cell line, Swiss 3T3, in the presence of SCF and IL-9, and investigated the characteristics of the cocultured mast cells. The presence of fibroblasts enhanced the induction of chymotryptic activity (Figure 2f). Carboxypeptidase A activities were significantly induced in the cocultured MCs/IL-9 (Figure 2h). Regarding the tryptic activity and histamine synthesis, the presence of fibroblasts reversed the actions of SCF and IL-9 (Figure 2g,i,j).

### 2.2. Antigen-Induced Degranulation Was Attenuated in MCs/IL-9 When They Were Sensitized with Lower Concentrations of IgE

Previous studies suggested that IL-9 alone should not be able to form a mast cell population from hematopoietic stem cells in the bone marrow but could support the expansion and maturation of a mucosal mast cell population [13]. We compared the characteristics of two model populations, BMMCs as a newly recruited immature population and MCs/IL-9 as a more differentiated mucosal population. MCs/IL-9 showed a similar profile to that of BMMCs in degranulation upon IgE-mediated antigen stimulation when they were sensitized with 1 µg/mL of IgE, whereas the levels of degranulation were significantly decreased in MCs/IL-9 when they were sensitized with 10 ng/mL IgE (Figure 3a,b). Decreases in the levels of degranulation were also found in MCs/IL-9 when they were stimulated with thapsigargin and A23187 (Figure 3c).

### 2.3. Degranulation of MCs/IL-9 in Response to ATP

Increased concentrations of ATP are often found in injured tissues and inflammatory sites. Several recent studies have indicated the significance of ATP-mediated activation of tissue mast cells [21]. Remarkable levels of degranulation (>50%) were found in BMMCs when they were stimulated with 0.3 mM ATP (Figure 4a). Such prominent responses were not observed in MCs/IL-9. We investigated the mRNA expression levels of several P_2_X receptor subtypes that have been previously reported to be expressed in mast cells [22]. Moderate and high levels of expression of *P2rx1*, *P2rx4* and *P2rx7* along with very low levels of expression of *P2rx3* were detected both in BMMCs and in MCs/IL-9 (Figure 4b,e). The expression levels of *P2rx7* were slightly, but not significantly, elevated in MCs/IL-9.

### 2.4. Induction of the Inflammatory Cytokine Gene Expression in MCs/IL-9

Mast cells are known to be the sources of a wide variety of cytokines. We compared the cytokine gene expression profiles between BMMCs and MCs/IL-9, considering IL-4, -5, -6, -13 and TNF-α (Figure 5). Because TNF-α was found to be pre-formed in murine mast cells [23], the release of TNF-α protein may not reflect the changes of mRNA expression. Overall, MCs/IL-9 were found to be poor producers of these cytokines, which play critical roles in various intestinal inflammatory responses, in comparison with BMMCs. MCs/IL-9 were found to be incapable of IL-4 synthesis. Lipopolysaccharide (LPS) could induce the expression of proinflammatory cytokines such as IL-6 and TNF-α in BMMCs, whereas it could only induce marginal levels of IL-6 in MCs/IL-9. MCs/IL-9 had potential to produce IL-5, IL-6, IL-13 and TNF-α upon IgE-mediated antigen stimulation.

## 3. Discussion

We characterized a novel bone-marrow-derived cultured mast cell model, which partly reflected the nature of newly recruited murine intestinal mast cells. Parasite infections or oral allergen-induced immediate responses cause drastic changes in the intestinal mucosa, whereas it remains largely unknown how local mast cells adapt themselves to these changes. We focused on the roles of SCF and IL-9, both of which were found to have major impacts on newly recruited intestinal mast cells. Accumulating evidence suggests that a large number of mast cells should be recruited into the intestinal tracts and that activated mast cells should play critical roles in pathology of inflammatory bowel diseases (IBD) and irritable bowel syndrome (IBS) [24,25,26,27,28], indicating that the characterization of intestinal mast cells should contribute to development of novel therapies for these chronic inflammatory diseases.

We previously found significant increases in histamine storage in BMMCs cocultured with Swiss 3T3 fibroblasts in the presence of SCF [18]. SCF alone was found to have a potential to induce histamine synthesis in BMMCs [29]. Here, histamine synthesis was abolished in the presence of SCF and IL-9, suggesting that IL-9 should be a potent suppressor of histamine synthesis. Histamine could profoundly affect the intestinal homeostasis through multiple pathways. It promotes increased vascular permeability and epithelial ion transport by acting on the H_1_ receptors [30,31] and recruits neutrophils by acting on the H_4_ receptors [32]. Forward et al. demonstrated in vitro that histamine could attenuate the suppressor functions of CD4^+^CD25^+^ T regulatory cells by acting on the H_1_ receptors [33]. IL-9 may adequately control the intestinal environment through limiting the effects of mast-cell-derived histamine. Interestingly, contact with fibroblasts was found to cancel the effects of IL-9. Migration of newly recruited mast cells in the intestinal tissues may stimulate histamine synthesis and promote further inflammatory responses (Figure 6).

IL-9 was found to induce the expression of *Mcpt1* and *Mcpt2* [13], which was often found during parasite infections and oral allergen-induced anaphylaxis [34,35]. Our findings are consistent with these findings. The gene-targeted mice lacking *Mcpt1* exhibited delayed expulsion of *Trichinella spiralis*, but not of *Nippostrongylus brasiliensis* [36]. Transcriptional induction of *Mcpt1* at the early phase should be required for efficient expulsion of *Trichinella spiralis*. Recently, mucosal mast cells were found to damage the epithelial barrier through release of Mcpt1 upon *Candida albicans* infection [37]. We noticed that tryptic activity was significantly decreased in the presence of SCF and IL-9, whereas this trend was entirely reversed by coculture with fibroblasts. This strict regulation of tryptase activity might reinforce the hypothesis that migration of recruited mast cells should regulate the intestinal epithelial integrity. In the jejunum of mice infected with *Trichinella spiralis*, Mcpt2^+^ mast cells were accumulated in the mucosal layer at the early phase, with Mcpt2^+^ and Mcpt6^+^ mast cells being detected at the recovery phase [35]. Mast cells recruited in the intestinal tissues upon parasite infection were often resident after its complete expulsion and were involved in pathogenic changes of the epithelial integrity. The gene-targeted mice lacking *Mcpt6* exhibited less severe pathology than the wild-type mice in sodium dextran sulfate and trinitrobenzene sulfonic acid (TNBS) induced colitis model [38], indicating that Mcpt6 should be involved in disruption of the epithelial integrity.

A variety of cells were found to be involved in recruiting mast cells through the release of IL-9. CD4^+^ IL-9 producing T cells (Th9) play critical roles in mast cell accumulation in allergic lung inflammation [39]. In the tolerant allografts, CD4^+^CD25^+^Foxp3^+^ regulatory T cells were found to recruit mast cells through production of IL-9 [40]. Interestingly, the mast cell itself was found to be the source of IL-9, which was augmented by lipopolysaccharide [41,42,43,44]. These findings raised a possibility that the number of intestinal mast cells should be drastically increased via the positive feedback loop. CD1d-restricted NKT cells, which are also the source of IL-9, were found to be involved in the recruitment of mast cell progenitors to the lung upon the aerosolized antigen challenge [45]. In this model, IL-9 might be involved in the early step of pulmonary mastocytosis.

We characterized the stimulated mediator release of MCs/IL-9 in comparison with BMMCs. BMMCs were found to be more sensitive to lower concentrations of IgE and the other secretagogues than MCs/IL-9. Kurashima et al. demonstrated that extracellular ATP should play critical roles in the pathology of TNBS-induced colitis model by acting on P_2_X_7_ purinoreceptors in intestinal mast cells [46]. In our system, ATP induced expression of IL-4 and IL-13 in BMMCs, whereas little or no induction of cytokine genes were observed in MCs/IL-9, raising a possibility that mast-cell-derived inflammatory cytokines should function at a very early stage of inflammation. Intestinal mast cells under the influence of IL-9 may contribute to inflammatory responses mainly through the release of chymases, but not histamine and proinflammatory cytokines.

We here established a novel culture model, which might be useful for investigation of spatiotemporal changes of the newly recruited intestinal mast cells. It should be required to integrate the findings obtained in vitro and in vivo for in-depth understanding of mastocytosis in the intestinal tissues and phenotypic changes of the intestinal mast cells.

## 4. Materials and Methods

### 4.1. Materials

The following materials were commercially obtained from the sources indicated: ATP, an anti-dinitrophenyl IgE antibody (clone SPE-7), p-nitrophenyl-β-d-2-acetoamide-2-deoxyglucopyranoside, probenecid and N-succinyl-Ala-Ala-Pro-Phe-pNA from Sigma-Aldrich (St. Louis, MO, USA), fetal bovine serum (FBS) from Thermo Fisher Scientific (Waltham, MA, USA), an anti-trinitrophenyl IgE antibody (clone IgE-3), an FITC-conjugated rat anti-mouse IgE antibody, a phycoerythrin-conjugated rat anti-mouse CD117 antibody and Fc blocker (clone 2.4G2) from BD Biosciences (San Diego, CA, USA), trinitrophenyl bovine serum albumin (TNP-BSA) from LSL (Tokyo, Japan), H-D-Ile-Pro-Arg-pNA (S-2288) from Chromogenix (Milano, Italy), *N*-(4-methoxyphenylazoformyl)-Phe-OH potassium salt (M-2245) from Bachem AG (Bubendorf, Switzerland), thapsigargin from Merck Millipore (Billerica, MA, USA), Fura-2/AM from Dojindo Laboratories (Kumamoto, Japan), recombinant mouse IL-9 from PeproTech (Rocky Hill, NJ, USA) and recombinant mouse IL-3 from R & D Systems (Minneapolis, MN, USA). Murine SCF was prepared by baculovirus expression system with Sf9 cells according to the procedure described previously [16]. All other chemicals were commercial products of reagent grade.

### 4.2. Animals

Specific-pathogen-free, 8- to 10-week-old male BALB/c mice were obtained from Japan SLC (Hamamatsu, Japan) and were kept in a specific-pathogen-free animal facility at Okayama University. This study was approved by the Committee on Animal Experiments of Okayama University (approved #OKU2012218 and #OKU2015040).

### 4.3. Preparation of Bone-Marrow-Derived Cultured Mast Cells

Preparation of IL-3-dependent bone-marrow-derived cultured mast cells (BMMCs) was performed as previously described [47]. Bone marrow cells obtained from male BALB/c mice were cultured in the presence of 10 ng/mL murine recombinant IL-3 for ~30 days. More than 95% of the cells exhibited metachromasy by acidic toluidine blue (pH 3.3) staining and were FcεRI^+^c-kit^+^ on the flow cytometry. IL-9-modified mast cells (MCs/IL-9) were prepared by the culture of BMMCs in the presence of 10 ng/mL IL-9 and 30 ng/mL SCF. Subculture was performed every other day for 12 days. In some experiments, BMMCs were cocultured with Swiss 3T3 cells, which were pretreated with mitomycin c as described previously [18], in the presence of 10 ng/mL IL-9 and 30 ng/mL SCF.

### 4.4. Flowcytometry

The cultured cells were treated with 10 µg/mL Fc blocker (clone 2.4G2) for 10 min at 4 °C and then further incubated with 12.5 µg/mL IgE (clone SPE-7) for 50 min at 4 °C. The surface expression levels of FcεRI and c-kit were measured using FACSCalibur (BD Biosciences, San Diego, CA, USA) with an FITC-conjugated rat anti-mouse IgE antibody and a phycoerythrin-conjugated rat anti-CD117 antibody.

### 4.5. Measurement of Granule Protease Activities

The enzymatic activities of three kinds of granule proteases, i.e., chymase, tryptase and carboxypeptidase A, were measured using their specific substrates as described previously [18]. The cells were washed in phosphate-buffered saline (PBS), incubated in PBS containing 2 M NaCl and 0.5% Triton X-100 at 4 °C for 30 min and then centrifuged at 12,000× *g* for 30 min at 4 °C to obtain the supernatant fractions. Chymotryptic activity was measured in 33 mM Tris-HCl, pH 8.3, containing 3.3 mM CaCl_2_ and 0.3 mM *N*-succinyl-Ala-Ala-Pro-Phe-*p*NA. Tryptic activity was measured in 33 mM Tris-HCl, pH 8.3, containing 2 mM S-2288. Carboxypeptidase A activity was measured in 33 mM Tris-HCl, pH 7.5, containing 0.6 mM M-2245. The enzymatic activity was calculated according to the changes of the values of OD_405_.

### 4.6. Measurement of Enzymatic Activity of Histidine Decarboxylase

The cultured cells were homogenized in the lysis buffer (10 mM potassium phosphate, pH 6.8 containing 10 mM KCl, 1.5 mM MgCl_2_, 1 mM EDTA, 1 mM EGTA, 0.2 mM dithiothreitol, 0.01 mM pyridoxal 5′-phosphate, 0.2 mM phenylmethylsulfonylfluoride, 0.1 mM benzamidine, 10 µg/mL aprotinin, 10 µg/mL leupeptin, 10 µg/mL E-64, 1 µg/mL pepstatin A and 0.1% Triton X-100) and centrifuged at 10,000× *g* for 30 min at 4 °C. The resultant supernatant was incubated in the assay buffer (0.1 M potassium phosphate, pH 6.8 containing 0.2 mM dithiothreitol, 0.01 mM pyridoxal 5′phosphate, 2% polyethylene glycol #300, 0.2 mM aminoguanidine and 0.8 mM L-histidine) at 37 °C for 4 h. The reaction was stopped by adding perchloric acid (3%). Histamine was fluorometrically measured by HPLC with a cation exchange column, WCX-1 (Shimadzu, Kyoto, Japan), after derivatization with *o*-phthalaldehyde [48].

### 4.7. Measurement of Degranulation

The cultured cells were suspended in PIPES-buffer (25 mM PIPES-NaOH, pH 7.4 containing 125 mM NaCl, 2.7 mM KCl, 1 mM CaCl_2_, 5.6 mM glucose and 0.1% bovine serum albumin) and then stimulated with the antigen or ATP for 30 min at 37 °C. They were centrifuged at 800× *g* at 4 °C for 5 min to obtain the supernatants (extracellular fractions, E). The resultant pellets were resuspended in PIPES-buffer containing 0.5% Triton X-100 and were centrifuged at 10,000× *g* for 10 min to obtain the supernatants (cell-associated fractions, C). The enzymatic activities of β-hexosaminidase were measured using the specific substrate, *p*-nitrophenyl-β-D-2-acetoamide-2-deoxyglucopyranoside (3.4 mM) in 67 mM citrate, pH 4.5. The amounts of *p*-nitrophenol were determined by measuring the values of OD_405_. The percentages of degranulation were calculated; E/(C + E) × 100 (%).

### 4.8. Quantitative RT-PCR

Total RNAs were extracted from the splenocytes using NucleoSpin RNA Kit (TaKaRa Bio Inc., Kusatsu, Japan) and reverse transcribed using PrimeScript^TM^ RT Reagent Kit (TaKaRa Bio Inc.). First strand DNAs were subjected to quantitative PCR using KOD SYBR qPCR Mix (TOYOBO, Osaka, Japan) or SYBR Green PCR Master Mix (Thermo Fisher Scientific) with the specific primer pairs as follows. *Mcpt1*: 5′-GCA CTT CTC TTG CCT TCT GG-3′, 5′-TAA GGA CGG GAG TGT GGT CT-3′; *Mcpt2*: 5′-GCA CTT CTT TTG CCT TCT GG-3′, 5′-TAA GGA CGG GAG TGT GGT TT-3′; *Mcpt5*: 5′-AGA ACT ACC TGT CGG C-3′, 5′-GTC GTG GAC AAC CAA AT-3′; *Cpa3*: 5′-GAT GTC TCG TGG GAC T-3′, 5′-GCC GTA GAT GTA ACG GG-3′; *Hdc*: 5′-TGC ACG CCT ACT ATC CTG CTC TTA C-3′, 5′-TCT GTG CAA GCT GGG CTA GAT G-3′; *Ptgr1*: 5′-CAT CGT GAA TCG GTG G-3′; *Il4*: 5′-TCG GCA TTT TGA ACG AGG TC-3′, 5′-GAA AAG CCC GAA AGA GTC TC-3′; *Il5*: 5′- ATG GAG ATT CCC ATG AGC AC-3′, 5′-GTC TCT CCT CGC CAC ACT TC-3′; *IL6*: 5′-TGG AGT CAC AGA AGG AGT GGC TAA G-3′, 5′-TCT GAC CAC AGT GAG GAA TGT CCA C-3′; *Il13*: 5′- CAG CTC CCT GGT TCT CTC AC-3′, 5′-CCA CAC TCC ATA CCA TGC TG-3′; *Tnf*: 5′-AAG CCT GTA GCC CAC GTC GTA-3′, 5′-GGC ACC ACT AGT TGG TTG TCT TTG-3′; *P2rx1*: 5′-CCA GGA CTT CCG AAG CCT TGC-3′, 5′-AGA ACT GTG GCC ACT CCA AAG ATG-3′; *P2rx3*: 5′-TCT TGC ACG AGA AGG CCT ACC AA-3′, 5′-GAT CTC ACA GGT CCG ACG GAC A-3′; *P2rx4*: 5′-GTG ACG TCA TAG TCC TCT ACT GT-3′, 5′-TGC TCG TAG TCT TCC ACA TAC TT-3′; *P2rx7*: 5′-AGC CTG TTA TCA GCT CCG TGC A-3′, 5′-TCA GGA CAC AGC GTC TGC ACT T-3′; *Gapdh*: 5′-TGT GTC CGT CGT GGA TCT GA-3′, 5′-TTG CTG TTG AAG TCG CAG GAG-3′.

### 4.9. Statistics

Statistical significance between two independent groups was determined by unpaired Student’s *t*-test. Statistical significance among multiple groups was determined using one-way ANOVA. Post-tests were performed with the Dunnett multiple comparison test for comparison with the control groups or the Tukey–Kramer multiple comparison test for all pairs of column comparison.

## Figures and Tables

**Figure 1 ijms-21-00236-f001:**
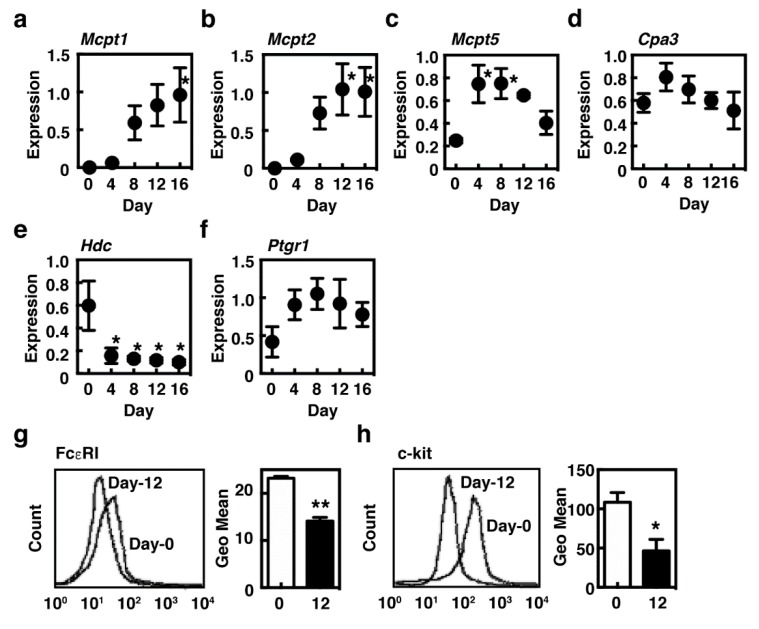
Induction of the characteristic genes of mucosal mast cells in the presence of interleukin-9 (IL-9) and stem cell factor (SCF). (**a**–**f**) Bone-marrow-derived cultured mast cells (BMMCs) were cultured in the presence of 10 ng/mL IL-9 and 30 ng/mL SCF for 16 days. Expression levels of mRNA of (**a**) *Mcpt1*, (**b**) *Mcpt2*, (**c**) *Mcpt5*, (**d**) *Cpa3*, (**e**) *Hdc* and (**f**) *Ptgr1* were measured by quantitative RT-PCR. The expression levels were normalized by measuring mRNA expression of *Gapdh*. The values are expressed as the means ± SEM (*n* = 3). Multiple comparisons were performed using one-way ANOVA with the Dunnett post-test. Values with * *p* < 0.05 are regarded as significant (vs. day 0). (**g**,**h**) Surface expression levels of (**g**) FcεRI and (**h**) c-kit of BMMCs (day 0, open columns) and MCs/IL-9 (day 12, closed columns) were measured by flow cytometry as described in Section 4. The mean fluorescence intensities are shown as the means ± SEM (*n* = 3, right panels). Statistical analysis was performed using Student’s *t*-test. Values with * *p* < 0.05 and ** *p* < 0.01 are regarded as significant.

**Figure 2 ijms-21-00236-f002:**
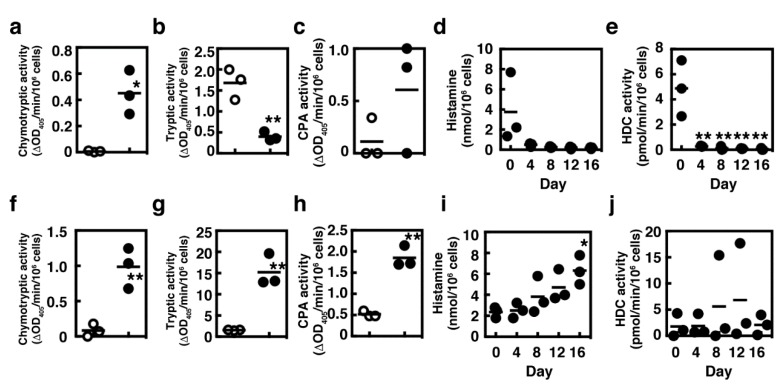
Effects of Swiss 3T3 fibroblastic cell line on the characteristic changes of the cultured mast cells in the presence of IL-9 and SCF. Enzymatic activities of (**a**,**f**) chymase, (**b**,**g**) tryptase and (**c**,**h**) carboxypeptidase A were measured in BMMCs (open circles, **a**–**c**,**f**–**h**) and in MCs cultured in the presence of IL-9 and SCF for 12 days (closed circles, **a**–**c**) and in MCs cocultured with Swiss 3T3 fibroblasts in the presence of IL-9 and SCF for 16 days (closed circles, **f**–**h**). Statistical analysis was performed using Student’s *t*-test. Values with * *p* < 0.05 and ** *p* < 0.01 are regarded as significant. (**d**,**i**) Cellular contents of histamine and (**e**,**j**) enzymatic activities of HDC were measured (**d**,**e**) in MCs cultured in the presence of IL-9 and SCF and (**i**,**j**) in MCs cocultured with Swiss 3T3 fibroblasts in the presence of IL-9 and SCF, respectively. Statistical analysis was performed using one-way ANOVA with Dunnett’s test. Values with * *p* < 0.05 and ** *p* < 0.01 are regarded as significant (compared with Day-0).

**Figure 3 ijms-21-00236-f003:**
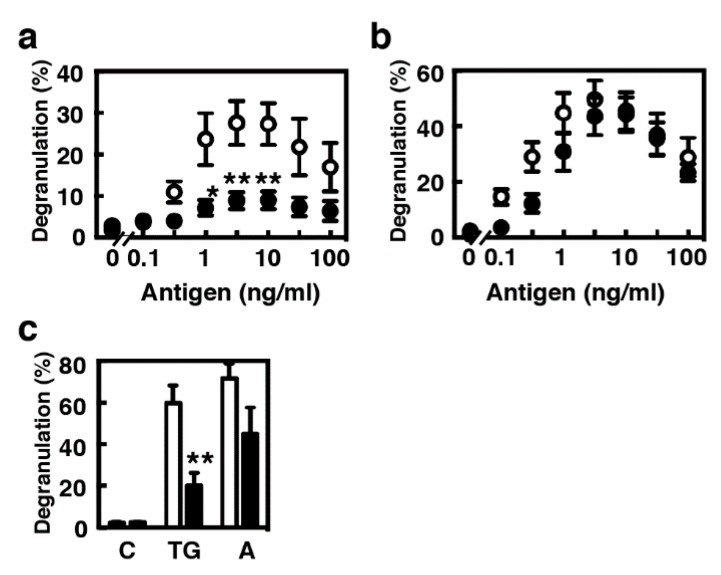
Comparison of the profiles of antigen-induced degranulation between BMMCs and MCs/IL-9. (**a**,**b**) BMMCs (open circles) or MCs/IL-9 (closed circles) were sensitized with (**a**) 10 ng/mL or (**b**) 1 µg/mL anti-TNP IgE (clone IgE-3) for 3 h at 37 °C. The cells were twice washed and stimulated with the indicated concentrations of the antigen (TNP-conjugated BSA). The levels of degranulation were determined by measuring the enzymatic activity of β-hexosaminidase. Values are presented as the means ± SEM (*n* = 4). Multiple comparisons were performed using two-way ANOVA with the Sidak test. Values with * *p* < 0.05 and ** *p* < 0.01 are regarded as significant. (**c**) BMMCs (open columns) or MCs/IL-9 (closed columns) were stimulated without (C, vehicle) or with 300 nM thapsigargin (TG) or 1 µM A23187 (A). The levels of degranulation were determined by measuring the enzymatic activity of β-hexosaminidase. Values are presented as the means ± SEM (*n* = 4). Statistical analysis was performed using Student’s *t*-test. A value with ** *p* < 0.01 is regarded as significant.

**Figure 4 ijms-21-00236-f004:**
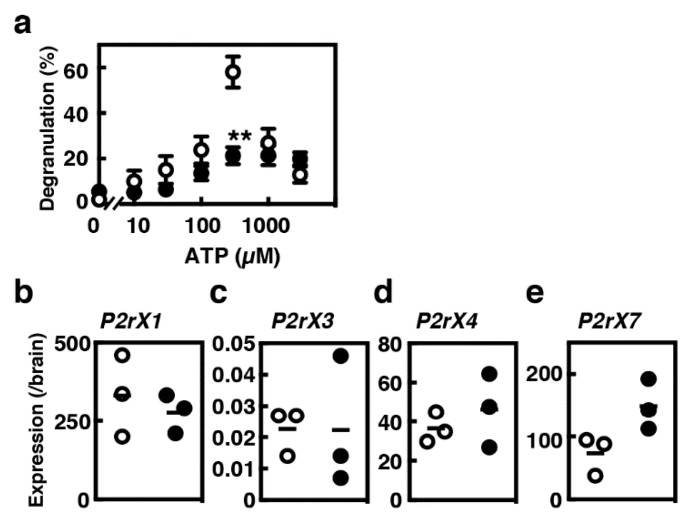
Comparison of the profiles of ATP-induced degranulation between BMMCs and MCs/IL-9. (**a**) BMMCs (open circles) and MCs/IL-9 (closed circles) were stimulated with the indicated concentrations of ATP. The levels of degranulation were determined by measuring the enzymatic activity of β-hexosaminidase. Values are presented as the means ± SEM (*n* = 4). Multiple comparisons were performed using two-way ANOVA with the Sidak test. A value with ** *p* < 0.01 is regarded as significant. (**b**–**e**) The levels of mRNA expression of (**b**) *P2rx1*, (**c**) *P2rx3*, (**d**) *P2rx4* and (**e**) *P2rx7* in BMMCs (open circles) and MCs/IL-9 (closed circles) were measured by quantitative RT-PCR.

**Figure 5 ijms-21-00236-f005:**
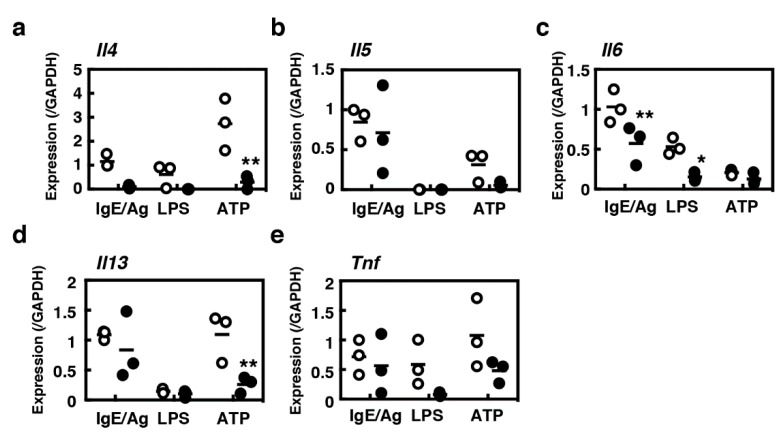
Comparison of transcriptional induction of various cytokines in between BMMCs and MCs/IL-9. BMMCs (open circles) or MCs/IL-9 (closed circles) were sensitized with 1 µg/mL anti-TNP IgE (clone IgE-3) for 3 h at 37 °C. The cells were twice washed and stimulated with the antigen (100 ng/mL, TNP-conjugated BSA) for 1 h (IgE/Ag). In parallel, BMMCs (open circles) or MCs/IL-9 (closed circles) were stimulated with LPS (1 µg/mL) or ATP (1 mM) for 1 h at 37 °C. Expression levels of mRNA of a series of cytokines ((**a**), IL-4; (**b**), IL-5; (**c**), IL-6; (**d**), IL-13; (**e**), TNF-α) were determined by quantitative RT-PCR. Statistical significance for comparison between BMMCs and MCs/IL-9 was determined by unpaired Student’s *t*-test. Values with * *p* < 0.05 and ** *p* < 0.01 are regarded as significant.

**Figure 6 ijms-21-00236-f006:**
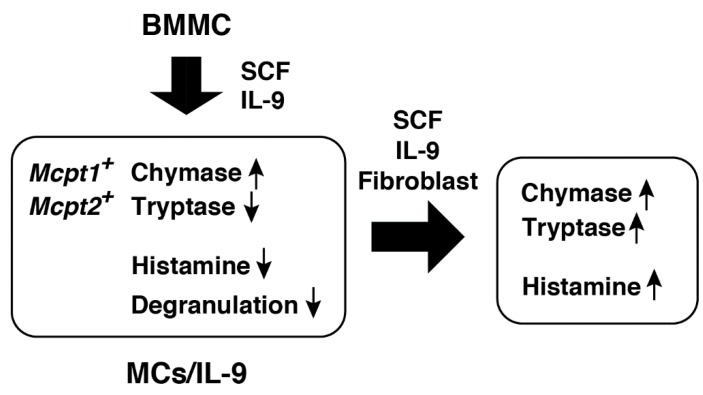
IL-9 has an impact on the characteristics of murine bone-marrow-derived cultured mast cells. The combination of SCF and IL-9 induced the expression of *Mcpt1* and *Mcpt2* in murine bone-marrow-derived cultured mast cells, as reported previously, and significantly suppressed the histamine synthesis and storage. IL-9-modified mast cells (MCs/IL-9) were less sensitive to ATP and the antigen in degranulation. The presence of fibroblasts recovered the histamine synthesis and tryptase expression.

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
