# Peer review of "Establishment and Characterization of a Murine Mucosal Mast Cell Culture Model"

_ijms, 2019, doi:10.3390/ijms21010236_

Round 1

Reviewer 1 Report

This is the first revision of a manuscript describing a cell culture model for obtaining cell populations in vitro that are similar to mucosal mast cells. For this purpose, IL-3-dependent bone marrow-derived mast cells were cultured in the presence of stem cell factor (SCF) and IL-9.

Overall reviewer comments were diligently followed and the manuscript is now much clearer, although a few problems persist, since apparently it is difficult to design a cell culture model of an insufficiently characterized cell population. Moreover the study is based on a comparatively small selection of characertistics and there are partly divergent changes with fibroblast co-culture. Unfortunately these changes do not yet concern the final Figure 6 that still lacks a proper Figure caption and description and uses the previous nomenclature. This Figure and legend still must be changed.

Author Response

I appreciate that the Reviewer would recognize that we could adequately address the comments. We will plan our next study according to the Reviewer's suggestion.

It is my regret that we did not change Figure 6 in context of the revision. We changed the title and graphic and added the text in the legend.

The Reviewer's input should strengthen our manuscript. I very much appreciate the Reviewer's support.

Reviewer 2 Report

Except for providing additional controls to demonstrate that the effects observed are due to IL-9 signaling in the mast cells rather than another variable, my concerns have been addressed by the revisions.

Author Response

I appreciate that the Reviewer would recognize that we could adequately address the comments. We have to refine the design to clarify the genuine effects of IL-9 on murine mast cells.

The Reviewer's input should strengthen our manuscript. I very much appreciate the Reviewer's support.

This manuscript is a resubmission of an earlier submission. The following is a list of the peer review reports and author responses from that submission.

Round 1

Reviewer 1 Report

For obtaining cell populations in vitro that are similar to mucosal mast cells, IL-3-dependent bone marrow-derived mast cells were cultured in the presence of stem cell factor (SCF) and IL-9.

1) The manuscript is complicated to read, because many facts that are important for the rationale are only explained in the discussion and other facts that are important to understand the course of the experiments are only found in the methods section at the end. Accordingly the introduction and results section must contain clear statements of the characteristics of mucosal mast cells, of the procedures with which the authors sought to obtain these features in cultured bone marrow-derived mast cells and to which degree these features were actually obtained.

2) The term “mucosal mast cell-like mast cells (MMC-like MCs) contains the term mast cells twice and therefore seems overly complicated. Please simplify.

3) All Figures lack captions that explain the main messages. For really appreciating the results it is important to recall the features of mucosal mast cells and to analyze, whether the cultivated mast cells also show them. Please design the presentation of the results accordingly.

Reviewer 2 Report

This manuscript by Kakinoki et al. characterizes mouse mast cells that have been differentiated in vitro via a novel protocol that extends the standard bone marrow mast cell (BMMC) culture protocol by culturing those BMMCs with IL-9 and SCF. Because IL-9 is critical in the accumulation of mucosal mast cells (MMCs) in response to helminth infection or allergic inflammation, the cells produced via this novel protocol could potentially be useful to study MMC differentiation and function in vitro or prepare such cells for reconstitution of MC- or MMC-deficient mice. However, despite some similarities, it is not clear that the MMC-like MCs produced via this protocol faithfully replicate features of MMCs, nor is it clear that IL-9 plays a role in the differentiation of MMCs in vivo. Without evidence indicating the physiological relevance of these cells or this differentiation pathway, the significance of this work is reduced. My individual comments can be found below.

Major:

It is possible that this in vitro differentiation protocol is entirely artificial and does not mimic the physiological differentiation of MMCs. For example, IL-9 has been reported to be important in the recruitment of mast cell progenitors to the lung as opposed to the differentiation of those cells to a mucosal mast cell phenotype (Jones, T., Hallgren, J., Humbles, A., et al. 2009. The Journal of Immunology 183, 5251-5260.) In addition, the preferential in vitro differentiation of MMCs has long been reported using IL-3 without SCF (Nakahata, T., Kobayashi, T., Ishiguro, A. et al. 1986. Nature 324, 65–67). Evidence showing that IL-9 is capable of causing MMC differentiation in vivo or further evidence demonstrating functional, phenotypic, or transcriptional similarities between these MMC-like MCs and MMCs would greatly strengthen this study. If, instead, the MMC-like MCs represent a distinct phenotype, this would also be of interest and should be explored and described in the manuscript. For the time course experiments in Figs. 1 and 2, a control treatment (such as SCF without IL-9) is necessary to attribute these changes to IL-9 signaling rather than, for example, a time-dependent phenomenon independent of IL-9 or due to removal of IL-3 from the culture. Figure 5 shows cytokine mRNA levels from BMMCs or MMC-like MCs in response to certain stimuli. However, neither protein levels nor the release of cytokines in response to these stimuli are explored. Given that some cytokines are pre-formed and need not be transcribed de novo, cytokine release is arguably the more relevant measure here.

Minor:

Transcript or protein names in the figures themselves would greatly facilitate the interpretation of the data (e.g., “Mcpt1 Expression” for Fig. 1a). Certain figure panels are not in order in the text, e.g. Figure 2g appears before 2d, and some panels are not cited in the text at all, such as Figure 2c and 2f. X-axis labels appear to be missing from Fig 3a and 3b. Some editing to improve the clarity and mechanics of the text would be helpful. For example, “Fog. 1e” on line 29 of page 2 should be “Fig. 1e” and “bacurovirus” on line 34 of page 7 should be “baculovirus.”